# Visceral and subcutaneous abdominal fat is associated with non-alcoholic fatty liver disease while augmenting Metabolic Syndrome's effect on non-alcoholic fatty liver disease: A cross-sectional study of NHANES 2017–2018

**Rebeca Garazi Elguezabal Rodelo**[1], **Leonardo M. Porchia**[1], **Enrique Torres-Rasgado**[1], **Esther López-Bayghen** [2]*, **M. Elba Gonzalez-Mejia**[1]*

1 Facultad de Medicina, Benemérita Universidad Autónoma de Puebla, Puebla, Puebla, México,
2 Departamento de Toxicología, Centro de Investigación y de Estudios Avanzados del Instituto Politécnico Nacional, Mexico City, Mexico

\* elba.gonzalez@correo.buap.mx (MEGM); ebayghen@cinvestav.mx (ELB)

## Abstract

### Background

The aim was to evaluate the effect different types of abdominal fat have on NAFLD development and the effects of abdominal fat has on the association between Metabolic Syndrome (MetS) and NALFD.

### Methods

Data was collected from the cross-sectional NHANES dataset (2017–2018 cycle). Using the controlled attenuation parameter (USG CAP, dB/m), which measures the level of steatosis, the cohort was stratified into two groups: NAFLD(+) ($\geq$274 dB/m) and NAFLD(-). Using complex samples analyses, associations between liver steatosis or NAFLD and types of abdominal fat area [Total abdominal (TAFA), subcutaneous (SAT), and visceral (VAT)] were determined. Pearson's correlation coefficient (r) was calculated to evaluate the associations between adipose tissues and NAFLD. Logistic regression was used to determine the risk [odds ratio (OR) and 95% confidence interval (95%CI)]. Participants were also classified by MetS, using the Harmonizing Definition criteria.

### Results

Using 1,980 participants (96,282,896 weighted), there was a significant (p<0.001) correlation between USG CAP and TAFA (r = 0.569), VAT (r = 0.645), and SAT (r = 0.479). Additionally, the risk of developing NAFLD was observed for total abdominal obesity (OR = 19.9, 95%CI: 5.1–77.8, p<0.001), visceral obesity (OR = 9.1, 95%CI: 6.2–13.5, p<0.001) and subcutaneous obesity (OR = 4.8, 95%CI: 3.2–6.9, p<0.001). Using 866 participants

**Data Availability Statement:** All data generated or analyzed during this study are available at the NHANES website (https://wwwn.cdc.gov/nchs/nhanes/default.aspx) and HAVARD (https://doi.org/10.7910/DVN/UW4EER).

**Funding:** This study was supported by grants from the Programa para el Desarrollo Profesional Docente (CA-160 FACMED) and the Vicerrectoría de Investigación, Benemérita Universidad Autónoma de Puebla, Mexico (GOMM-SAL22-G). The funders had no role in the study's design, data collection or analysis, decision to publish, or preparation of the manuscript.

(44,399,696 weighted), for visceral obesity, participants with MetS and visceral obesity (OR = 18.1, 95%CI: 8.0–41.3, p<0.001) were shown to have a greater risk than participants with MetS only (OR = 6.3, 95%CI: 2.6–15.2, p<0.001). For subcutaneous obesity, again, participants with MetS and subcutaneous obesity (OR = 18.3, 95%CI: 8.0–41.9, p<0.001) were shown to have a greater risk than the MetS-only group (OR = 10.3, 95%CI: 4.8–22.4, p<0.001).

## Conclusion

TAFA, VAT, and SAT were positively associated with USG CAP values and increased the risk of developing NAFLD. Also, the type of abdominal fat depots did affect the association between MetS and NAFLD.

## Introduction

Non-alcoholic fatty liver disease (NAFLD) is the leading cause of chronic hepatic disease in the World and now affects 32.4% of the population [1]. The highest prevalence is seen in the Middle East and South American countries, and the lowest is in African countries, suggesting different ethnic-based mechanisms that affect NAFLD development [2, 3]. NAFLD is defined as the accumulation of fat in >5% of the hepatocytes with a clinical exclusion of significant alcohol consumption [4]. NAFLD physiopathology is not fully understood; regardless, it can be attributed to the interaction of several risk factors and comorbidities, such as genetic susceptibility, gut dysbiosis, obesity, and Metabolic Syndrome (MetS) [3, 5].

Obesity is defined by the World Health Organization as an "excessive fat accumulation" and is clinically classified with the body mass index (BMI), despite BMI's known inability to differentiate between lean and fat tissues [6, 7]. Traditionally, it was believed that adipose tissue was a single, functional, uniform organ that passively responded to certain stimuli. Now, it is known that adipose tissue is metabolically active and that not all fat depots are equally hazardous for health [8]. Fat in different anatomical locations, known as "adipose tissue distribution," has an important role in the properties and functions of different types of adipose tissue [9]. When adipose tissue is stored primarily in the abdominal region, its accumulation is called "central or abdominal obesity," which is recognized as a risk factor for metabolic diseases. The accumulation of fat in the abdomen can be divided into two main compartments: subcutaneous adipose tissue (SAT) and visceral adipose tissue (VAT) [10]. VAT is different from SAT when concerned with venous drainage. VAT drains directly into the portal vein, whereas venous drainage for SAT is directly into the circulatory system [11]. Therefore, metabolic products of VAT reach the liver directly, and for SAT, they are systemic [10, 12].

Changes in fat distribution, increased accumulation of VAT, and/or impaired SAT function can reflect the dysfunctional capacity of adipose tissue to respond to metabolic demands [13]. This impaired adipose tissue is now known as "adiposopathy," which literally translates to sick adipose tissue [14]. Healthy adipose tissue is characterized by its ability to expand and increase the number of adipocytes in response to an energy surplus. Normally, this expansion is overseen by SAT. However, when the lipid handling capacity of SAT is inadequate to the amount of positive caloric intake, this favors VAT expansion. In the visceral depot, these expanded adipocytes are in a hyperlipolytic state and insensitive to insulin, favoring lipid overspill towards the liver [15]. Hence, the accumulation of VAT directly impacts the accumulation of fat in the liver.

Initially, MetS was proposed as a combination of components (central obesity, high blood pressure, dyslipidemia, and hyperglycemia) that increased the risk of cardiovascular events. Nevertheless, secondary outcomes for the presence of those same factors have been identified, such as NAFLD, which is considered the hepatic manifestation of MetS [16]. Patients with MetS are 11 times more likely to develop NAFLD than their metabolically healthy counterparts [17]. Out of the components for MetS, obesity is recognized as the main risk factor for NAFLD. The current definition of MetS uses waist circumference (WC) to identify the presence of central obesity; however, this index fails to differentiate VAT and SAT. In women, augmented VAT and decreased SAT were associated with MetS, whereas in men, augmented SAT was shown to have adverse effects with each MetS component [18]. Meanwhile, in a longitudinal study, VAT area was correlated with increased MetS incidence, while SAT was shown to have a protective effect [19]. Consequently, some studies posit that the difference in the accumulation of VAT and SAT can affect the association between MetS and NAFLD.

Even though the association between NAFLD and obesity is undeniable, little is known about how different types of adipose tissue influence NAFLD development, and large-scale cohort studies assessing this association are needed. Here, using the data from the National Health and Nutritional Examination Survey (NHANES), the effect of abdominal adipose tissue on NAFLD prevalence was evaluated. Additionally, the influence that ethnicity has on the disease was explored. Lastly, the effect of VAT and SAT on NAFLD in the presence of MetS was evaluated.

## Materials and methods

### Data source

The National Center for Health Statistics (NCHS) at the Centers for Disease Control and Prevention conducts the NHANES, a large cross-sectional survey that systematically gathers data on medical examinations, laboratory testing, and interviews for studying a range of variables of medical importance [20]. The survey consists of dietary, laboratory, body measurement examination, interview, and demographic informatics, collected from a multi-stage stratified probability design in a sample population that oversamples certain population groups to obtain a nationally representative sample of civilians in the United States of America. Before collection of the data, informed consent was obtained from each participant. Data gathering was approved by the Ethics Review Board for the NCHS, and files were posted online for public use [21]. NHANES provides a full description of data collection procedures and methods [22]. The NCHS Research Ethics Review Board approved the NHANES study protocols (Protocol #2011–17; Protocol #2018–01). The Center for Disease Control and Prevention conducted the survey, and all participants reviewed and signed a comprehensive informed consent. All procedures performed in this study involving human participants were conducted in accordance with the ethical standards of the institutional and/or national research committee and with the 1964 Helsinki Declaration and its later amendments. All data generated or analyzed during this study are available at the NHANES website (https://wwwn.cdc.gov/nchs/nhanes/default.aspx) and HARVARD [23].

### Study population

A defined set of eligibility criteria was constructed according to the patient population, intervention, comparison group, outcomes, and study design (PICOS) question scheme. The PICOS question was: In participants with NAFLD, do different types of abdominal adipose tissue, when compared to participants without NAFLD, increases the prevalence and risk of developing NAFLD, as determined using the NHANES cross-sectional dataset? The eligibility

criteria reflected the PICOS components and the subsequent inclusion and exclusion criteria. Therefore, the primary outcome was to evaluate how different types of abdominal adipose tissue affect liver steatosis and NAFLD. Since different ethnic groups are shown to have different distributions of abdominal adipose tissues, secondary outcomes were to examine ethnic effects on liver steatosis and NAFLD. Lastly, an alternative secondary outcome was the effect of different distributions of abdominal adipose tissues have on the MetS/NALFD interaction.

In this study, the data from the 2017–2018 cycle was used. To be included, the participants had to be 1) non-pregnant females or males, aged $\geq$18 years, 2) BMI $\geq$18.5 kg/m$^2$, 3) had results for Dual-Energy X-Ray Absorptiometry (DEXA) for the abdominal area, and 4) had results for a hepatic ultrasound with the "controlled attenuation parameter" (USG CAP). They were excluded for 1) liver diseases other than NAFLD (Hepatitis B/C/D, autoimmune, or hepatocarcinoma), 2) significant alcohol consumption (30 g/day for males and 20 g/day for females), 3) participants with HIV, and 4) partial USG CAP exams.

## Measurement methods and instrumentation

**Demographic data.** Key demographic variables collected were age, biological sex, and ethnicity. The age (years) was determined at the interview. Biological sex was described as either male or female. Ethnicity was categorized by the NHANES as Non-Hispanic White, Mexican American, Other Hispanics, Non-Hispanic Black, Non-Hispanic Asian, or Other Races (including multiracial).

**Anthropometric variables.** Anthropometric variables [weight (WT, kg), height (HT, m), BMI (kg/m$^2$), WC (cm), systolic and diastolic blood pressures (SBP and DBP, respectively, mmHg)] were collected according to a standardized protocol [24]. BMI was categorized into Normal weight (18.5–24.9 kg/m$^2$), Overweight (25–29.9 kg/m$^2$), Obese Class I (30–34.9 kg/m$^2$), Obese Class II (35–39.9 kg/m$^2$), Obese Class III (>40 kg/m$^2$), according to the World Health Organization criteria [6].

**Laboratory values.** Fasting plasma glucose (FPG, mg/dL), insulin (FPI, μU/mL), glycated hemoglobin (HbA1c, %), total cholesterol (TC, mg/dL), high-density lipoprotein (HDL, mg/dL), low-density lipoprotein (LDL, mg/dL), triglycerides (TG, mg/dL), aspartate aminotransferase (AST, U/L), alanine aminotransferase (ALT, U/L), platelets (10$^3$ cells/mL), and ferritin (ng/mL) were analyzed according to a standardized protocol [24]. Insulin resistance was calculated according to the Homeostatic Model Assessment for Insulin Resistance (HOMA1-IR) equation: (FPG x FPI)/405. A score $\geq$2.5 were considered positive for insulin resistance [25].

**Body fat composition.** Body fat composition was collected at a NHANES mobile examination center during the medical examination. Total abdominal fat area (TAFA, cm$^2$), VAT (cm$^2$), and SAT (cm$^2$), as well as android and gynoid fat mass (g), were determined by DEXA. Visceral obesity (VATob) was categorized using VAT with a cutoff value of 100 cm$^2$ into VATob or visceral lean [26]. Subcutaneous obesity (SATob) was categorized into two groups, elevated subcutaneous fat or low subcutaneous fat, using the median of SAT as the cutoff value. Total abdominal obesity (TAFAob) was categorized using TAFA as elevated abdominal fat and low abdominal fat, using a cutoff value of 130 cm$^2$ [27].

**Liver steatosis.** Liver steatosis was assessed using USG CAP, a noninvasive method for detecting hepatic steatosis based on transient elastography, as indicated by Sasso *et al.* [28]. The USG CAP value (dB/m) was categorized according to Karlas *et al.* into S0 (<248 dB/m), S1 (248–268 dB/m), S2 (268–280 dB/m), and S3 (>280 dB/m) [29]. The study population was divided into two groups according to their NAFLD status using the USG CAP value as NAFLD(+), $\geq$274 dB/m, and NAFLD(-), <274 dB/m [30].

**Metabolic syndrome.** The Harmonizing definition was used to classify subjects as normal [MetS(-)] or having MetS [MetS(+)]. The Harmonizing Definition [31] requires three of the following five criteria: 1) WC: ≥90 cm for men or ≥80 cm for women; 2) TG ≥150mg/dL; 3) HDL <40 mg/dL for men or <50 mg/dL for women; 4) SBP ≥130 mmHg or DBP ≥85 mmHg; or 5) FPG ≥100mg/dL.

## Statistical analyses

All analyses were carried out with the Statistical Package for the Social Sciences software v26.0 (SPSS, IBM Corp., Armonk, NY, USA) using either sample weights or the complex samples study design option. The mean or percentage with standard errors were calculated for the quantitative variables and categorical variables, respectively. Normality of continuous variables was determined using the Kolmogorov–Smirnov test. The differences between the groups were determined by the Rao Scott-Chi$^2$ test for categorical data, whereas the Complex Samples General Linear Model was used for continuous data. Receiver-operating characteristic (ROC) curve analysis was used to determine sensitivity and specificity between NAFLD and TAFA, SAT, VAT, and BMI. The area under the ROC curve (AUC) was calculated using the method described by Hanley and McNeil [32]. Using sensitivity and specificity, Youden's index (sensitivity + specificity– 1) was calculated and the highest score was considered the optimal cutoff value. Complex Samples Logistic Regression was performed to calculate the odds ratio (OR) with a 95% confidence interval (95%CI). The Pearson correlation coefficient (r) was calculated to assess the presence of the correlation, and linear regression was used to calculate the beta coefficients and 95%CIs to evaluate the effect TAFA, VAT, SAT, and BMI have on the components of MetS and USG CAP. Comparisons between the correlation coefficients were determined by calculating Steiger's Z [33]. The Jockheere-Terpstra test determined a trend between USG CAP and the number of MetS components. P-values <0.05 (two-tailed) were considered statistically significant.

The NHANES does not collect a complete set of data for each participant; therefore, certain variables may have missing data. The analyses were carried out in 2 stages: 1) the complete cohort to determine the effect VAT and SAT have on hepatic steatosis and NAFLD and 2) a MetS sub-analysis. For the complete cohort, patients were included if data was present for USG CAP, TAFA, VAT, and SAT. Other variables were shown only if <5% of the data was missing. When selecting variables to be adjusted, if the sample size decreased by >5% as well as the portions between the independent and dependent variable changed by >1%, then the variable would not be included. To be included for the MetS sub-analysis, the participant had to have no missing data for USG CAP, TAFA, VAT, and SAT as well as WC, SBP, DBP, HDL, TG, and FPG.

To adjust any associations between the independent variables and USG CAP/NAFLD, univariate logistic regression was conducted using baseline characteristics variables to identify potential confounder variables for NAFLD. Afterward, a multivariable logistic regression model was first built by including all statistically significant variables from the univariate analysis. Then, non-significant variables were sequentially removed. Key variables (biological sex, age, and ethnicity) remained in the final model, independent of whether the p-value was significant, as these variables are known to affect the risk and prevalence of both MetS and NAFLD.

## Results

### Selection of participants

From the NHANES 2017–2018 cycle, 9,254 participants were available; however, 56.2% were excluded due to not having DEXA results available, and an additional 9.6% were removed due

to an incomplete or partial USG CAP result. Of the 3,167 remaining participants, 37.5% were excluded due to age, BMI, significant alcohol consumption, liver disease, HIV, or Hepatitis. Ultimately, the cohort consisted of 1,980 participants (96,282,896 weighted). Of these participants, for the MetS sub-analysis, only 866 (44,399,696 weighted) had data for WC, SBP, DBP, TG, HDL, and FPG. Details of the selection process of the participants are shown in the S1 Fig.

## Characteristics of the population

The characteristics of the studied population are presented in Table 1. Concerning the total population, males and females were equally distributed (50.9% vs 49.1%, respectively). The prevailing ethnicity present in the cohort was Non-Hispanic Whites (55.7%), followed by Non-Hispanic Blacks (12.5%). When the cohort was stratified by NAFLD status, 36.0% were diagnosed with NAFLD. When the two groups were compared, there were more males in the NAFLD(+) group; moreover, the NAFLD(+) group was older and had, as expected, worse values in parameters associated with hyperglycemia (HbA1c), dyslipidemia (TC and HDL), liver function (ALT, platelets, and Ferritin), and obesity (WC, BMI, TAFA, VAT, and SAT). Interestingly when stratified by BMI class, a majority of the NAFLD(-) group was normal weight or overweight (79.2%), whereas a majority of the NAFLD(+) group was obese (69.1%). Similar results were observed with the MetS sub-set, especially with the distribution for ethnicity, biological sex, and BMI categories, as well as with laboratory results for glycemic parameters, lipid profile, and liver parameters. Here, 34.2% were diagnosed as NAFLD(+). Insulin resistance was higher in the NAFLD(+) group (73.7%); however, 33.0% of the NAFLD(-) group was determined to have insulin resistance. As expected, each component for MetS was significantly worse in the NAFLD(+) group than in the NAFLD(-) group. The VATob and SATob rates were 2.8 and 2.0 times higher in the NAFLD(+) group, respectively.

## Visceral fat correlates better with USG CAP

TAFA, VAT, SAT, and BMI were plotted against USG CAP scores (Fig 1), and the association was evaluated. TAFA (r = 0.569, p<0.001), VAT (r = 0.645, p<0.001), SAT (r = 0.479, p<0.001), and BMI (r = 0.580, p<0.001) were strongly correlated with hepatic steatosis. However, when the type of adipose tissue was compared, VAT was more strongly correlated than TAFA, SAT, and BMI for USG CAP ($p_{comparison}$<0.001, S1 Table). The association was confirmed with linear regression. The beta coefficients were significant for TAFA, VAT, SAT, and BMI, even after controlling for age, biological sex, ethnicity, ALT, HbA1c, and HDL (Table 2).

ROC analysis was used to compare the effectiveness of the type of abdominal fat and BMI for determining NAFLD (Fig 2). VAT had the higher AUC for NAFLD (AUC = 0.83, 95%CI: 0.81–0.85, p<0.001), followed by BMI (AUC = 0.80, 95%CI: 0.78–0.82, p<0.001), TAFA (AUC = 0.78, 95%CI: 0.76–0.80, p<0.001), and SAT (AUC = 0.74, 95%CI: 0.72–0.76, p<0.001). When each AUC was compared, VAT's AUC was superior ($p_{comparison}$<0.05, S2 Table), suggesting that measuring VAT would be a better index for NAFLD. Nevertheless, using the highest Youden index, cutoffs for VAT, BMI, TAFA, and SAT were determined to be ≥102.7 cm$^2$ (Youden index = 0.507, specificity = 79.0%, sensitivity = 71.7%), ≥27.2 kg/m$^2$ (Youden index = 0.465, specificity = 62.9%, sensitivity = 83.6%), ≥408.7 cm$^2$ (Youden index = 0.422, specificity = 65.3%, sensitivity = 76.8%), and ≥322.0 cm$^2$ (Youden index = 0.350, specificity = 64.8%, sensitivity = 70.2%), respectively.

## The risk of developing NAFLD by type of abdominal fat

Univariate logistic regression was performed for all baseline variables to identify potential confounding variables for NAFLD (Table 3). Age, biological sex, ethnicity, BMI, HbA1c, TC,

**Table 1. Characteristics of the cohort.**

| Categories | Complete cohort | | | MetS sub-analysis | | |
|---|---|---|---|---|---|---|
| | NAFLD(-) [a] | NAFLD(+) [a] | p-value [b] | NAFLD(-) [a] | NAFLD(+) [a] | p-value [b] |
| N-unweighted | 1255 | 725 | | 566 | 300 | |
| N-weighted | 61,649,532 | 34,633,364 | | 29,172,074 | 15,227,622 | |
| Biological sex (male %) | 46.5 ± 2.1 | 58.9 ± 2.5 | 0.001* | 49.7 ± 2.5 | 59.8 ± 3.1 | 0.005* |
| Ethnicity (%) | | | 0.004* | | | 0.004* |
| Mexican-American | 8.4 ± 1.7 | 14.9 ± 3.5 | | 8.1 ± 1.8 | 16.0 ± 4.8 | |
| Other Hispanic | 9.3 ± 1.4 | 7.8 ± 1.4 | | 9.0 ± 1.8 | 6.3 ± 1.8 | |
| Non-Hispanic White | 55.9 ± 3.1 | 55.3 ± 3.9 | | 56.6 ± 4.1 | 60.5 ± 3.8 | |
| Non-Hispanic Black | 13.9 ± 2.0 | 9.9 ± 2.3 | | 13.3 ± 2.6 | 8.1 ± 1.8 | |
| Non-Hispanic Asian | 7.9 ± 1.4 | 7.1 ± 1.4 | | 6.8 ± 1.4 | 5.9 ± 1.2 | |
| Other Race | 4.7 ± 0.8 | 5.0 ± 1.1 | | 6.1 ± 1.4 | 3.3 ± 1.0 | |
| *Anthropometric variables* | | | | | | |
| Age (years) | 36.0 ± 0.6 | 41.7 ± 0.5 | <0.001* | 35.7 ± 0.7 | 41.7 ± 0.6 | <0.001* |
| Weight (kg) | 74.8 ± 0.8 | 96.2 ± 0.9 | <0.001* | 75.3 ± 1.4 | 95.5 ± 1.4 | <0.001* |
| Height (cm) | 168.1 ± 0.3 | 169.3 ± 0.5 | 0.017* | 168.7 ± 0.3 | 169.8 ± 0.5 | 0.097 |
| BMI (kg/m$^2$) | 26.4 ± 0.3 | 33.5 ± 0.3 | <0.001* | 26.4 ± 0.5 | 33.1 ± 0.5 | <0.001* |
| Normal weight (%) | 45.8 ± 2.7 | 4.7 ± 1.2 | <0.001* | 49.2 ± 4.2 | 6.1 ± 2.2 | <0.001* |
| Overweight (%) | 33.4 ± 2.5 | 26.1 ± 2.8 | | 29.6 ± 3.2 | 24.4 ± 3.3 | |
| Obese I (%) | 14.6 ± 2.0 | 33.9 ± 2.7 | | 13.6 ± 2.9 | 36.5 ± 4.1 | |
| Obese II (%) | 3.9 ± 0.7 | 21.1 ± 1.6 | | 4.3 ± 0.9 | 21.5 ± 3.9 | |
| Obese III (%) | 2.3 ± 0.6 | 14.1 ± 1.8 | | 3.3 ± 1.3 | 11.5 ± 1.8 | |
| WC (cm) | 90.4 ± 0.7 | 109.3 ± 0.8 | <0.001* | 90.6 ± 1.2 | 109.0 ± 1.2 | <0.001* |
| SBP (mmHg) | VNS [c] | VNS [c] | | 116.0 ± 1.0 | 123.8 ± 0.9 | <0.001* |
| DBP (mmHg) | VNS [c] | VNS [c] | | 71.3 ± 1.0 | 76.0 ± 0.8 | 0.001* |
| *Glycemic profile* | | | | | | |
| FPG (mg/dL) | VNS [c] | VNS [c] | | 100.3 ± 0.6 | 116.9 ± 3.2 | <0.001* |
| HbA1c (%) | 5.3 ± 0.1 | 5.8 ± 0.1 | <0.001* | 5.3 ± 0.1 | 5.8 ± 0.1 | <0.001* |
| FPI (μU/dL) | VNS [c] | VNS [c] | | 9.5 ± 0.5 | 18.9 ± 1.4 | <0.001* |
| HOMA1-IR | VNS [c] | VNS [c] | | 2.4 ± 0.2 | 5.9 ± 0.6 | <0.001* |
| IR(+) (%) | VNS [c] | VNS [c] | | 33.0 ± 3.4 | 73.7 ± 4.2 | <0.001* |
| *Lipid profile* | | | | | | |
| TC (mg/dL) | 183.8 ± 2.1 | 193.7 ± 2.4 | <0.001* | 181.6 ± 2.8 | 191.0 ± 4.5 | 0.120 |
| HDL (mg/dL) | 55.7 ± 0.6 | 46.0 ± 0.5 | <0.001* | 55.4 ± 0.8 | 46.7 ± 0.8 | <0.001* |
| LDL (mg/dL) | VNS [c] | VNS [c] | | 108.9 ± 2.2 | 116.0 ± 3.1 | 0.118 |
| TG (mg/dL) | VNS [c] | VNS [c] | | 86.2 ± 2.5 | 145.2 ± 10.3 | <0.001 |
| *Liver parameters* | | | | | | |
| AST (U/L) | VNS [c] | VNS [c] | | 20.5 ± 0.6 | 22.8 ± 0.8 | 0.043* |
| ALT (U/L) | 19.7 ± 0.3 | 29.0 ± 0.7 | <0.001* | 19.5 ± 0.5 | 29.1 ± 1.1 | <0.001* |
| Platelets (10$^3$ cells/μL) | 244.9 ± 2.7 | 252.9 ± 3.4 | 0.037* | 238.1 ± 3.7 | 247.0 ± 3.7 | 0.059 |
| Ferritin (ng/mL) | 113.5 ± 7.2 | 172.6 ± 9.7 | <0.001* | 114.2 ± 8.6 | 171.5 ± 12.1 | 0.002* |
| USG CAP | 217.5 ± 1.3 | 322.3 ± 2.0 | <0.001* | 218.0 ± 1.6 | 320.2 ± 2.9 | <0.001* |
| *Abdominal Fat Distribution* | | | | | | |
| Android fat mass (g) | 1886 ± 60 | 3366 ± 67 | <0.001* | 1915 ± 106 | 3318 ± 115 | <0.001* |
| Gynoid fat mass (g) | 4230 ± 91 | 5481 ± 113 | <0.001* | 4257 ± 155 | 5407 ± 117 | <0.001* |
| TAFA (cm$^2$) | 360.0 ± 10.7 | 573.2 ± 10.7 | <0.001* | 360.0 ± 18.5 | 566.9 ± 15.5 | <0.001 |
| TAFAob (%) | 95.4 ± 0.7 | 93.0 ± 1.0 | | 92.2 ± 1.8 | 99.7 ± 0.3 | <0.001* |
| SAT (cm$^2$) | 284.4 ± 8.5 | 434.2 ± 10.2 | <0.001* | 282.4 ± 15.4 | 429.3 ± 13.3 | <0.001* |

*(Continued)*

**Table 1.** (Continued)

| Categories | Complete cohort | | | MetS sub-analysis | | |
|---|---|---|---|---|---|---|
| | NAFLD(-) [a] | NAFLD(+) [a] | p-value [b] | NAFLD(-) [a] | NAFLD(+) [a] | p-value [b] |
| SATob (%) | 37.7 ± 2.7 | 74.2 ± 2.5 | | 37.0 ± 5.1 | 74.9 ± 3.5 | <0.001* |
| VAT (cm$^2$) | 75.6 ± 2.6 | 139.0 ± 2.1 | <0.001* | 77.6 ± 3.7 | 137.6 ± 4.3 | <0.001* |
| VATob (%) | 24.1 ± 2.5 | 74.4 ± 2.2 | | 27.0 ± 3.4 | 75.0 ± 3.7 | |
| *Metabolic Syndrome* | | | | | | |
| Prevalence (%) | - | - | | 14.7 ± 2.1 | 51.1 ± 4.2 | <0.001* |
| Central Obesity (%) | - | - | | 37.9 ± 3.3 | 86.8 ± 2.8 | <0.001* |
| Hypertension (%) | - | - | | 20.0 ± 3.1 | 37.9 ± 2.8 | 0.003* |
| Hypertriglyceridemia (%) | - | - | | 12.2 ± 1.5 | 35.1 ± 5.4 | <0.001* |
| Low HDL (%) | - | - | | 11.9 ± 2.2 | 26.4 ± 4.3 | 0.011* |
| Hyperglyceridemia (%) | - | - | | 38.7 ± 2.9 | 71.8 ± 3.3 | <0.001* |

Abbreviations: ALT: alanine transaminase; AST: aspartate aminotransferase; BMI: Body-mass index; DBP: diastolic blood pressure; FPG: fasting plasma glucose; FPI: fasting insulin; HbA1c: glycated hemoglobin; HDL: high-density lipoprotein; HOMA1-IR: Homeostatic Model Assessment for Insulin Resistance; IR(+): subjects with insulin resistance; LDL: low-density lipoprotein; NAFLD(-): subjects without Non-alcoholic Fatty Liver Disease; NAFLD(+): subjects with Non-alcoholic Fatty Liver Disease; SAT: subcutaneous adipose tissue; SATob: subcutaneous obesity; SBP: systolic blood pressure; TAFA: total abdominal fat area; TAFAob: total abdominal obesity; TC: total cholesterol; TG: triglycerides; WC: waist circumference; USG CAP: ultrasound with the Controlled Attenuation Parameter; VAT: visceral adipose tissue; VATob: visceral obesity; and VNS: Value not shown.

[a] Data is presented in mean or frequency ± standard error.

[b] p-value corresponds to the difference between NAFLD(-) and NAFLD(+) determined by the Roa-Scott Chi2 test or the Complex Sample Design General Linear Model. * indicates statistically significant results (p<0.05, two-tailed)

[c] Value was not reported due to >5% of participants missing the value.

HDL, ALT, platelets, and ferritin were initially identified as potential confounders. Using multivariate logistic regression, age, ethnicity, BMI, HbA1c, HDL, and ALT were shown to retain their associations after including all variables. In the end, the "selected" variables included age, biological sex (due to the published studies indicating biological sex as a factor for NAFLD development), ethnicity (due to Mexican Americans consistently demonstrating an increased risk), HbA1c, HDL, and ALT. BMI was excluded from the "selected" model because it was controlled for by stratification.

For the type of abdominal fat, TAFAob, VATob, and SATob were associated with NAFLD development (Table 4). Interestingly, only for VATob, when stratified by BMI classification, the group that presented with the highest OR was the normal-weight participants, followed by Obese Class II, Overweight, and Obese Class I. Interestingly, obese class III showed no significant association. When adjusted by age, biological sex, ethnicity, ALT, HbA1c, and HDL, and when BMI categorization was not considered, the association between TAFAob, VATob, and SATob and the development of NAFLD remained. However, the effect was lost when BMI class was considered for VATob. For SATob, the "selected" model showed an effect for the normal weight group.

## The effect of ethnicity on NAFLD

When the cohort was stratified by ethnicity and when the steatosis classification was considered, Mexican Americans presented with more S1-S3 subjects than any other ethnicity (Fig 3). Moreover, Mexican Americans presented with the highest rate of NAFLD(+) subjects (49.9%), greater than Other Hispanics (32.0%), Non-Hispanic Whites (35.7%), Non-Hispanic Blacks (28.5%), Non-Hispanic Asians (33.7%), and Other Races (37.6%). This correlated with an

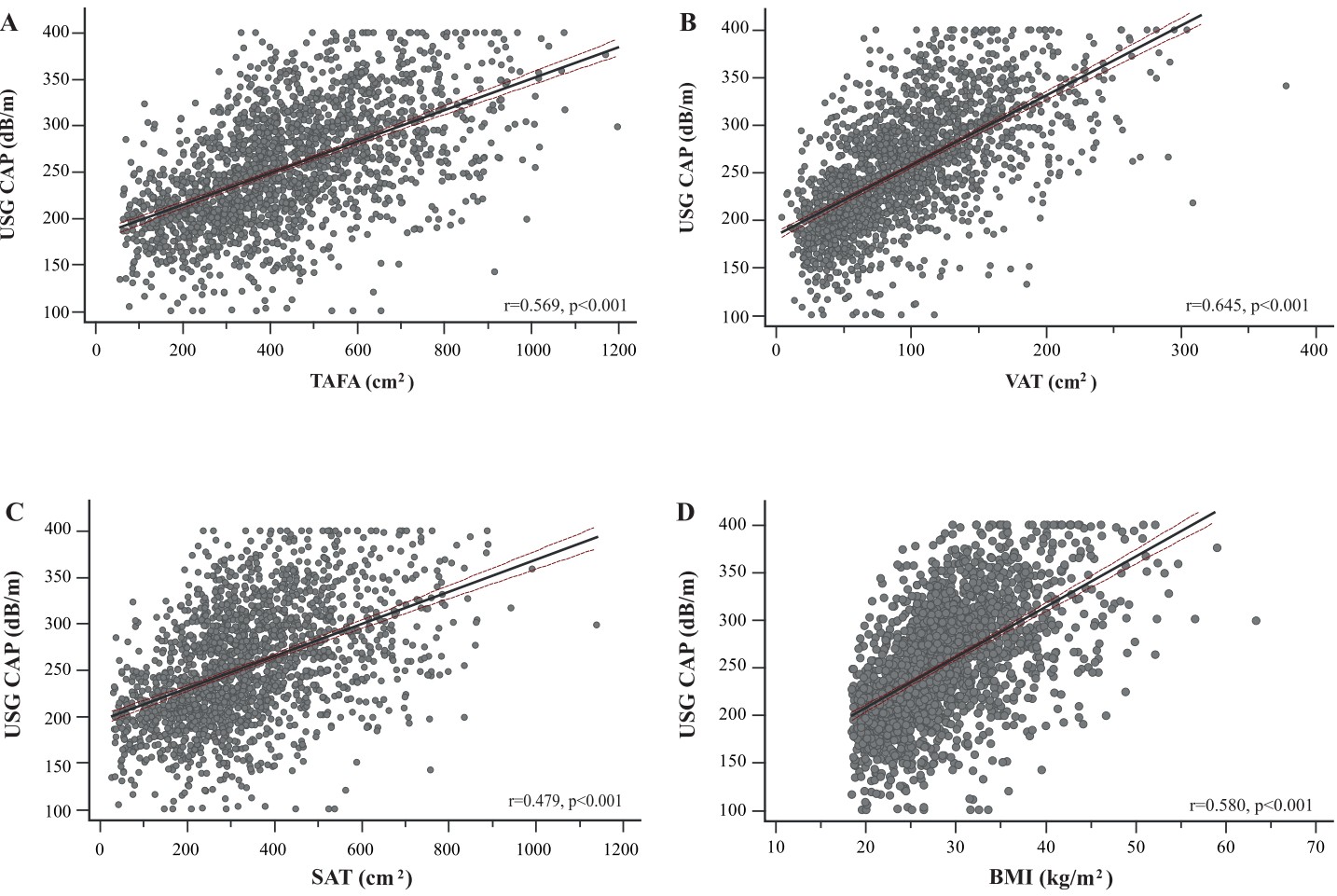

**Fig 1. Association between different types of abdominal fat and the controlled attention parameter.** Scattergrams were constructed in which **A** total abdominal fat area (TAFA), **B** visceral adipose tissue (VAT), **C** subcutaneous adipose tissue (SAT), and **D** body-mass index (BMI) were compared to liver elastography controlled attenuation parameter (USG CAP) values. Pearson's correlation coefficient was calculated to determine the strength of the association. Lines correspond to the linear fit with 95% confidence intervals.

**Table 2. Linear regression between USG CAP and type of abdominal fat depots and BMI.**

| Category | Crude [a] | Model 1 [b] | Model 2 [c] |
|---|---|---|---|
| BMI | 5.4 (4.7–6.1), p<0.001* | 5.2 (4.6–5.8), p<0.001* | 4.2 (3.4–5.1), p<0.001* |
| TAFA | 0.17 (0.15–0.19), p<0.001* | 0.18 (0.17–0.20), p<0.001* | 0.15 (0.13–0.17), p<0.001* |
| VAT | 0.71 (0.64–0.79), p<0.001* | 0.71 (0.64–0.77), p<0.001* | 0.56 (0.46–0.65), p<0.001* |
| SAT | 0.18 (0.15–0.20), p<0.001* | 0.21 (0.19–0.23), p<0.001* | 0.17 (0.15–0.20), p<0.001* |

Abbreviations: ALT: alanine transaminase; BMI, body mass index; HbA1c: glycated hemoglobin; HDL: high-density lipoprotein; SAT, subcutaneous adipose tissue; TAFA, total abdominal fat area; USG CAP: ultrasound with the Controlled Attenuation Parameter; and VAT, visceral adipose tissue.

[a] Values are the beta coefficients (95% confidence interval), and p-value. Values were calculated using the Complex Sample Design General Linear Model. * Indicates a significant result (p<0.05, two-tailed).

[b] Model 1 = crude model adjusted for age and sex.

[c] Model 2 = crude model adjusted for age, sex, ethnicity, ALT, HbA1c, and HDL.

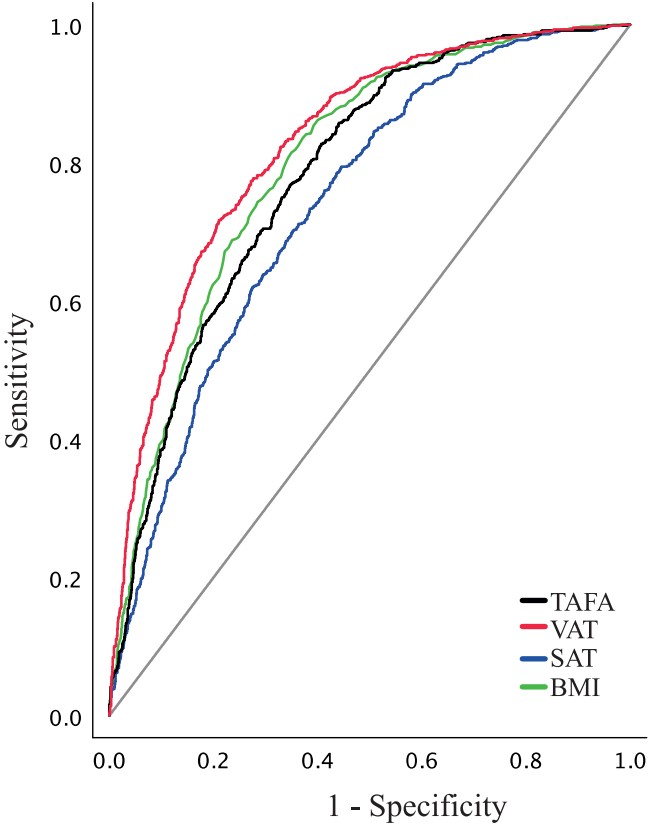

**Fig 2. The diagnostic capability of different types of abdominal fat and BMI to determine NAFLD.** Receiver operating characteristic curves were constructed, and the area under the curve (AUC) was calculated for the TAFA (black line), VAT (red line), SAT (blue line), and BMI (green line) to determine NALFD. The reference line (diagonal line) corresponds to no predictability (AUC = 0.50).

elevated risk of developing NAFLD for Mexican Americans compared to Non-Hispanic Whites (Table 3). Non-Hispanic Blacks, which had the lowest prevalence of NAFLD, had a decreased risk of developing NAFLD; however, when VATob was considered within each ethnicity, Non-Hispanic Blacks had an increased risk (Table 5). Moreover, Non-Hispanic Whites had the highest risk of developing NAFLD, followed by Non-Hispanic Asians. Interestingly, Mexican Americans presented with a lower risk than the other ethnic groups. When adjusted by the "selected" model, the association was lost for Mexican Americans; however, Non-Hispanic Whites and Non-Hispanic Blacks still had the highest risk for NAFLD. For SATob, each ethnicity presented with similar levels of risk for NAFLD; however, Other Hispanics, Non-Hispanic Blacks, and Non-Hispanic Whites presented with higher ORs than Non-Hispanic Asians and Mexican Americans. The observation remained even after adjusting for age, biological sex, ALT, HbA1c, and HDL.

## VATob and SATob augmented the risk that MetS conferred on NAFLD development

Few studies have shown that MetS can augment NAFLD development; therefore, the effect of VATob and SATob was assessed. Using the MetS sub-set, Mets components were plotted against USG CAP scores (Fig 4). WC strongly correlated with liver steatosis (r = 0.635,

**Table 3. Univariate and multivariate logistic regression analysis for the presence of NAFLD.**

| Variables | N-weighted [a] | Univariate [b] | Multivariate [b] | Selected [b] |
|---|---|---|---|---|
| Age (per 10 years) | 96,282,896 (36.0%) | 1.5 (1.4–1.6), <0.001* | 1.4 (1.2–1.6), <0.001* | 1.4 (1.3–1.6), <0.001* |
| Biological sex | | | | |
| Male | 49,027,202 (41.6%) | 1.0 Referent | 1.0 Referent | 1.0 Referent |
| Female | 47,255,692 (30.2%) | 0.6 (0.5–0.8), 0.001* | 0.9 (0.6–1.3), 0.510 | 1.2 (0.9–1.6), 0.265 |
| Ethnicity | | | | |
| Non-Hispanic White | 53,594,086 (35.7%) | 1.0 Referent | 1.0 Referent | 1.0 Referent |
| Mexican-American | 10,363,545 (49.9%) | 1.8 (1.3–2.5), 0.003* | 1.8 (1.0–3.3), 0.042* | 1.7 (1.0–2.9), 0.035* |
| Other Hispanic | 8,415,657 (32.0%) | 0.8 (0.6–1.3), 0.401 | 0.9 (0.6–1.5), 0.782 | 0.8 (0.4–1.3), 0.293 |
| Non-Hispanic Black | 11,996,434 (28.5%) | 0.7 (0.5–1.0), 0.048* | 0.5 (0.3–0.9), 0.028* | 0.7 (0.5–1.1), 0.132 |
| Non-Hispanic Asian | 7,313,128 (33.7%) | 0.9 (0.6–1.4), 0.648 | 1.6 (1.1–2.3), 0.027* | 0.8 (0.5–1.3), 0.303 |
| Other Races | 4,600,047 (37.6%) | 1.1 (0.6–1.9), 0.762 | 0.7 (0.2–2.2), 0.548 | 0.9 (0.4–2.2), 0.829 |
| BMI (per 5 kg/m$^2$) | 96,282,896 (36.0%) | 2.9 (2.4–3.6), <0.001* | 2.6 (2.0–3.3), <0.001* | - |
| HbA1c (per 0.5%) | 92,676,057 (36.4%) | 1.8 (1.4–2.3), <0.001* | 1.3 (1.1–1.6), 0.005* | 1.5 (1.2–1.8), <0.001* |
| TC (per 10 mg/dL) | 92,039,455 (36.6%) | 1.1 (1.0–1.1), 0.006* | 1.0 (1.0–1.1), 0.205 | - |
| HDL (per 10 mg/dL) | 92,039,455 (36.6%) | 0.5 (0.5–0.6), <0.001* | 0.7 (0.6–0.8), <0.001* | 0.6 (0.5–0.6), <0.001* |
| ALT (per 10 mg/dL) | 91,698,075 (36.6%) | 1.6 (1.4–1.7), <0.001* | 1.3 (1.1–1.4), <0.001* | 1.4 (1.2–1.5), <0.001* |
| Platelets (per 100 x 10$^3$ cells/mL) | 93,126,337 (36.6%) | 1.3 (1.0–1.5), 0.039* | 1.0 (0.7–1.4), 0.984 | - |
| Ferritin (per 100 mg/dL) | 92,247,021 (36.6%) | 1.4 (1.1–1.7), 0.004* | 1.1 (0.9–1.3), 0.603 | - |

Abbreviations: ALT: alanine transaminase; BMI: Body-mass index; HbA1c: glycated hemoglobin; HDL: high-density lipoprotein; NAFLD: non-alcoholic Fatty Liver Disease; and TC: total cholesterol.

[a] Total weighted number of participants (percent positive for NAFLD).

[b] Values are the odds ratio (95% confidence interval), and p-values. Values were calculated using Complex Sample Design Logistic Regression. * Indicates a significant result (p<0.05, two-tailed).

p<0.001), while SBP (r = 0.323, p<0.001), DBP (r = 0.296, p<0.001), TG (r = 0.432, p<0.001), HDL (r = -0.364, p<0.001), and FPG (r = 0.312, p<0.001) were moderately correlated. There was a positive trend between the number of MetS components and USG CAP levels (p$_{Jockheere-Terpstra}$<0.001). Again, linear regression confirmed the association between components of MetS and USG CAP, even after controlling for age, biological sex, ethnicity, ALT, and HbA1c (Table 6). Interestingly, the presence of MetS was associated with a 60.4 dB/m increase, 36.6 dB/m after adjusting. When the cohort was stratified by MetS and VATob, there was a similar increase in risk for the VATob-only group (MetS-/VATob+; OR = 7.3) and the MetS-only group (MetS+/VATob-; OR = 6.3, Table 7). However, when both conditions were present (MetS+/VATob+), there was a 2.5- to 2.9-increase in the risk. When controlled for age, biological sex, ethnicity, ALT, and HbA1c, the results remained significant. For SATob, similar observations were observed. The OR for the MetS-only group (MetS+/SATob-; OR = 10.3) was almost 2-fold higher than the OR for the SATob-only group (MetS-/SATob+; OR = 5.7). Again, the highest OR was when both conditions were present (MetS+/SATob+, OR = 18.3). Interestingly, these results were significantly affected when controlling for age, biological sex, ethnicity, ALT, and HbA1c. Overall, this suggests a potential biological interaction between different types of abdominal adipose tissue and MetS, augmenting the development of NAFLD.

## Discussion

Using the 2017–2018 NHANES dataset, the association between different types of abdominal fat depots and NAFLD was evaluated, in which TAFA, VAT, and SAT were positively

**Table 4. The risk associated with the type of abdominal fat depots for NAFLD, stratified by BMI.**

| Type of abdominal obesity | Referent group [a] | Obese group [b] | Crude [c] | Model 1 [d] | Model 2 [e] |
|---|---|---|---|---|---|
| *TAFAob* | | | | | |
| Overall | 4,451,999 (2.9%) | 91,830,897 (37.6%) | 19.9 (5.1–77.8), <0.001* | 17.9 (4.6–70.1) <0.001* | 6.6 (1.3–33.5) 0.025* |
| Normal weight | 4,058,734 (2.7%) | 25,788,528 (5.9%) | 2.3 (0.4–14.4), 0.361 | 1.5 (0.2–9.3), 0.648 | 1.0 (0.1–10.2), 0.966 |
| Overweight | 393,265 (5.3%) | 29,218,945 (30.8%) | 8.0 (0.8–85.9), 0.081 | 9.1 (0.7–112.6) 0.082 | 2.5 (0.2–27.2) 0.423 |
| Obese Class I [f] | 0 (0.0%) | 20,776,652 (56.6%) | NA | NA | NA |
| Obese Class II [f] | 0 (0.0%) | 9,745,396 (75.1%) | NA | NA | NA |
| Obese Class III [f] | 0 (0.0%) | 6,301,378 (77.6%) | NA | NA | NA |
| *VATob* | | | | | |
| Overall | 55,645,495 (15.9%) | 40,637,402 (63.4%) | 9.1 (6.2–13.5), <0.001* | 8.4 (5.5–13.0), <0.001* | 4.4 (2.6–7.4), <0.001* |
| Normal weight | 27,997,587 (4.1%) | 1,849,675 (26.1%) | 8.2 (2.4–28.0), 0.002* | 3.5 (0.9–13.0), 0.060 | 2.3 (0.6–9.5), 0.220 |
| Overweight | 17,722,419 (20.1%) | 11,889,792 (45.9%) | 3.4 (1.5–7.4), 0.005* | 2.5 (1.0–6.2), 0.058 | 1.3 (0.4–4.0), 0.663 |
| Obese Class I | 7,596,896 (38.5%) | 13,179,756 (67.0%) | 3.3 (1.7–6.2), 0.001* | 3.5 (1.6–7.7), 0.004* | 2.2 (0.9–5.7), 0.087 |
| Obese Class II | 1,859,283 (48.0%) | 7,886,112 (81.4%) | 4.8 (2.1–10.8) 0.001* | 3.1 (1.1–8.5), 0.029* | 1.6 (0.6–4.8), 0.342 |
| Obese Class III | 469,311 (71.2%) | 5,823,068 (78.2%) | 1.4 (0.3–6.0), 0.558 | 1.0 (0.3–4.1), 0.972 | 0.4 (0.1–1.9), 0.206 |
| *SATob* | | | | | |
| Overall | 47,387,494 (18.9%) | 48,895,402 (52.5%) | 4.8 (3.2–6.9), <0.001* | 8.3 (5.3–12.9), <0.001* | 6.3 (4.1–9.6), <0.001* |
| Normal weight | 26,894,996 (5.0%) | 2,952,267 (10.4%) | 2.2 (0.7–7.1), 0.157 | 3.8 (1.2–11.7), 0.025* | 5.1 (1.8–14.9), 0.005* |
| Overweight | 16,131,093 (32.2%) | 13,481,118 (28.5%) | 0.8 (0.4–1.6), 0.582 | 1.6 (0.8–3.6) 0.195 | 1.6 (0.7–3.7) 0.277 |
| Obese Class I | 3,919,672 (50.8%) | 16,856,980 (57.9%) | 1.3 (0.6–3.0), 0.453 | 1.9 (0.7–4.8), 0.180 | 2.4 (0.9–6.0), 0.063 |
| Obese Class II [f] | 441,735 (100%) | 9,303,661 (73.9%) | NA | NA | NA |
| Obese Class III [f] | 0 (0.0%) | 6,301,378 (77.6%) | NA | NA | NA |

Abbreviations: ALT: alanine transaminase; BMI: Body-mass index; HDL: high-density lipoprotein; NA: not applicable; NAFLD: subjects without Non-alcoholic Fatty Liver Disease; SATob: subcutaneous obesity; TAFAob: total abdominal obesity; and VATob: visceral obesity.

[a] Total weighted number of participants of the referent group (percent positive for NAFLD). The referent group was low abdominal fat for TAFAob, visceral lean for VATob, and low subcutaneous fat for SATob.

[b] Total weighted number of participants of the obese group (percent positive for NAFLD). The obese group was elevated fat for TAFAob, visceral obesity for VATob, and elevated subcutaneous fat for SATob.

[c] Values are odds ratios (95% confidence interval), and p-value. Values were determined by comparing the obese group to the referent group for NAFLD using Complex Samples Design Logistic Regression. * Indicates a significant result (p<0.05, two-tailed).

[d] Model 1 = crude model adjusted for age and sex

[e] Model 2 = crude model adjusted for age, sex, ethnicity, ALT, HbA1c, and HDL.

[f] The ORs could not be calculated due to 0 or 100% values.

correlated with the development of NAFLD. Even though VAT was more associated with USG CAP and a better predictor for NAFLD, TAFA showed a greater risk for NAFLD development. Concerning MetS, its components were identified as determinant factors that individually and collectively increased the risk of developing NAFLD; nevertheless, this association was affected by the accumulation of adipose tissue in different abdominal fat depots.

It is widely recognized that obesity is a major risk factor for the development of NAFLD [34], which is typically measured by clinicians with BMI [35, 36]. This method could overestimate the risk for people with high BMIs and low-fat mass and, at the same time, underestimate the risk for subjects with normal BMIs and high-fat mass [37], especially if VAT and SAT proportions are not considered. BMI presented with a moderate correlation with USG CAP; however, VAT's correlations were statistically better than BMI as well as TAFA and SAT. This was expected as for VAT releases adipokines, TGs, and free fatty acids into the portal vein, which directly leads to the liver [38]. Other studies also confirm that VAT promotes liver steatosis

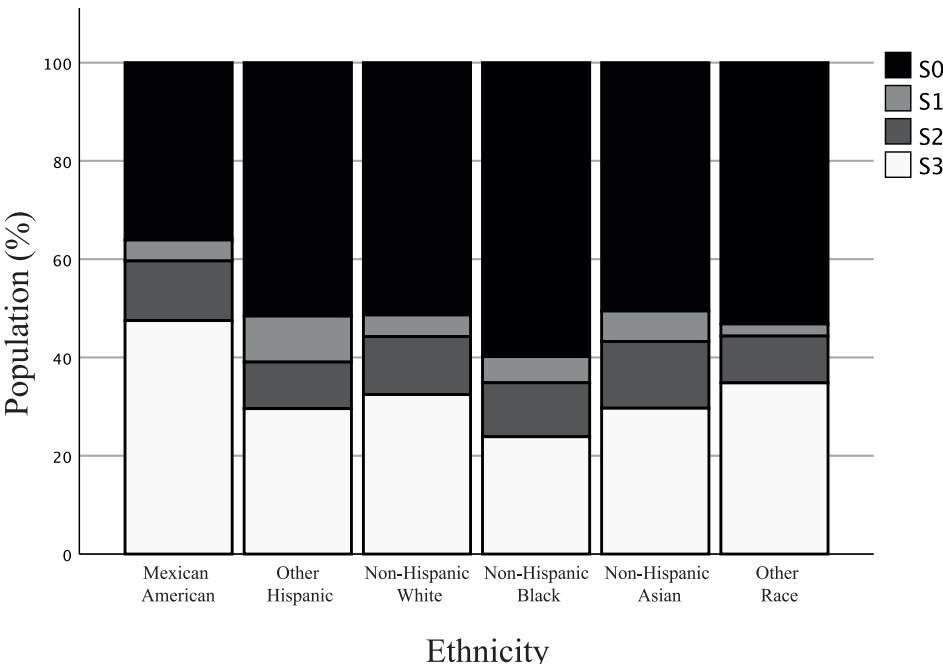

**Fig 3. Distribution of steatosis severity when categorized by ethnicity.** Using the controlled attenuation parameter, steatosis was stratified as S0 (<248 dB/m, black), S1 (248–268 dB/m, light grey), S2 (268–280 dB/m, dark grey), and S3 (>280 dB/m, white).

more than SAT and is a better maker than BMI [39, 40]. Using linear regression, a 1 unit increase in BMI was associated with a 5.4 dB/m increase in USG CAP, which appears to be better than VAT (1 cm$^2$ augmented USG CAP by 0.71 dB/m). However, due to the ranges associated with BMI and VAT, it is posited that VAT could account more for the USG CAP range than BMI. In support of this, VAT was determined to be a better predictor of NAFLD than BMI, SAT, and TAFA. Interestingly, the cutoff for VAT was 102.7 cm$^2$, which is close to the definition for VATob. For TAFA, the cutoff was 408.7 cm$^2$, which is well above the accepted 130 cm$^2$ threshold, suggesting that the current criteria overestimates obesity and risk for NAFLD. Concerning SAT, there is no accepted cutoff; therefore, we are proposing using 322.0 cm$^2$ for future studies.

Different types of obesity are known to affect certain diseases differently, such as Type 2 Diabetes, cardiovascular diseases, dyslipidemias, etc [41]. Here, the results suggest that subjects with NAFLD have a higher propensity for elevated WC and increased TAFA, indicating a tendency toward central obesity. Indeed, there was a prominent association between TAFAob and NAFLD risk (OR = 19.9), which remained after adjusting for key variables. However, neither TAFA nor WC can differentiate between different quantities of VAT and SAT [42]. Here, VATob (OR = 9.1) and SATob (OR = 4.8) are also associated with NAFLD risk. The importance of differentiating between these two depots of abdominal adipose tissue is based on their different metabolic phenotypes and their consequences [43]. Different factors regulate adipose tissue accumulation in different depots [44]. The accumulation of VAT directly affects the physiopathology of NAFLD mainly due to 1) an increased lipolysis that generates free fatty acids, 2) insulin resistance, and 3) the production of several adipokines resulting in a proinflammatory state, all of which are associated with NAFLD progression and severity [38]. Stefan *et al.* demonstrated that increased visceral and liver fat depots, as well as low leg fat mass,

**Table 5. The risk associated with VATob and SATob for NAFLD, stratified by ethnicity.**

| Ethnicity | Referent group [a] | Obese group [b] | Crude [c] | Model 1 [d] | Model 2 [e] |
|---|---|---|---|---|---|
| *VATob* | | | | | |
| Non-Hispanic White | 29,433,976 (12.5%) | 24,161,010 (64.0%) | 12.5 (5.9–26.2), <0.001 * | 10.8 (5.0–23.5), <0.001 * | 5.3 (2.2–12.6), 0.001 * |
| Mexican-American | 5,052,623 (30.2%) | 5,310,922 (68.6%) | 5.1 (2.0–12.9), 0.002 * | 5.7 (1.6–20.1), 0.010 * | 2.5 (0.6–10.4), 0.171 |
| Other Hispanic | 5,178,423 (16.8%) | 3,237,235 (56.5%) | 6.4 (2.8–14.9), <0.001 * | 6.5 (2.9–14.6), <0.001 * | 4.0 (1.5–10.7), 0.009 * |
| Non-Hispanic Black | 8,588,725 (14.6%) | 3,407,710 (63.7%) | 10.3 (6.2–17.2), <0.001 * | 9.6 (5.5–16.9), <0.001 * | 5.8 (2.8–11.9), <0.001 * |
| Non-Hispanic Asian | 4,829,909 (19.2%) | 2,483,220 (62.1%) | 6.9 (3.7–12.7), <0.001 * | 6.0 (2.5–14.4), 0.001 * | 3.1 (1.3–7.7), 0.018 * |
| Other | 2,562,741 (24.4%) | 2,037,306 (54.2%) | 3.7 (1.4–9.7), 0.012 * | 3.0 (1.0–9.4), 0.057 | 1.3 (0.5–3.7), 0.600 |
| *SATob* | | | | | |
| Non-Hispanic White | 25,973,600 (18.7%) | 27,620,486 (51.8%) | 4.7 (2.5–8.9), <0.001 * | 9.1 (4.1–20.3), <0.001 * | 6.6 (2.9–15.3), <0.001 * |
| Mexican-American | 4,061,433 (30.9%) | 6,302,112 (62.1%) | 3.6 (1.5–9.1), 0.008 * | 6.7 (2.6–17.3), 0.001 * | 4.2 (1.4–13.0), 0.017 * |
| Other Hispanic | 4,812,525 (15.4%) | 3,603,133 (54.3%) | 6.5 (3.9–11.1), <0.001 * | 8.5 (4.8–14.8), <0.001 * | 7.1 (3.5–14.5), <0.001 * |
| Non-Hispanic Black | 5,765,896 (12.4%) | 6,230,539 (43.5%) | 5.5 (2.9–10.1), <0.001 * | 7.8 (4.4–14.1), <0.001 * | 6.5 (2.6–16.5), 0.001 * |
| Non-Hispanic Asian | 4,876,331 (24.3%) | 2,436,797 (52.6%) | 3.5 (2.5–4.9), <0.001 * | 6.1 (4.5–8.2), <0.001 * | 4.1 (2.3–7.2), <0.001 * |
| Other | 1,897,710 (11.3%) | 2,702,337 (56.1%) | 10.1 (2.7–37.6), 0.002 * | 14.5 (4.0–52.9), 0.001 * | 15.1 (4.3–52.3), <0.001 * |

Abbreviations: ALT: alanine transaminase; BMI: Body-mass index; HDL: high-density lipoprotein; NA: not applicable; NAFLD: subjects without Non-alcoholic Fatty Liver Disease; SATob: subcutaneous obesity; TAFAob: total abdominal obesity; and VATob: visceral obesity.

[a] Total weighted number of participants of the referent group (percent positive for NAFLD). The referent group was visceral lean for VATob and low subcutaneous fat for SATob.

[b] Total weighted number of participants of the obese group (percent positive for NAFLD). The obese group was visceral obese for VATob and elevated subcutaneous fat for SATob.

[c] Values are odds ratios (95% confidence interval), and p-value. Odds ratios were determined by comparing the obese group to the referent group for NAFLD using Complex Samples Design Logistic Regression. * indicates a significant result (p<0.05, two-tailed).

[d] Model 1 = crude model adjusted for age and biological sex.

[e] Model 2 = crude model adjusted for age, biological sex, ethnicity, ALT, HbA1c, and HDL.

might be the result of impaired expandability of healthy SAT stores, resulting in adiposopathy [45]. Adiposopathy is the combination of the accumulation of VAT and ectopic fat, inflammation, impaired adipose tissue expandability and adipogenesis, hypertrophy, and altered lipid metabolism [14]. In other studies, VATob has been shown to significantly increase the development of NAFLD [46], which is in accordance with our results. In fact, subjects with NAFLD were classified as VATob 2.1-times more than subjects without NAFLD, a difference greater than the observed for TAFAob and SATob. Nevertheless, when adjusting for age, biological sex, HDL, HbA1c, and ALT, this risk decreased. This could be because, as mentioned before, the accumulation of fat in the liver attributed to VAT increased lipolysis and insulin resistance —fundamental factors directly associated with HbA1c and HDL.

NAFLD in normal-weight individuals, also known as lean-NAFLD, presents in 7–20% of all NAFLD cases [47]. Here, the rate was lower (6.1%) and depending on the type of abdominal fat, TAFA (2.7%), VAT (4.1%), or SAT (5.0%), the effect on NAFLD risk was modified. In this study, when the cohort was stratified by BMI categories, in the unadjusted model, the highest risk for NAFLD was seen for lean participants with VATob, contrary to the expected results— normal-weight individuals are considered metabolically healthy. There was no effect for TAFAob and SATob. Interestingly, the risk decreased as the BMI categories increased, becoming non-significant in patients defined as Obese Class III. Recently, this contradiction has been investigated, in which normal-weight patients with NAFLD might have worse outcomes than their obese counterparts [48]. To date, this contradiction is not fully understood but it can be

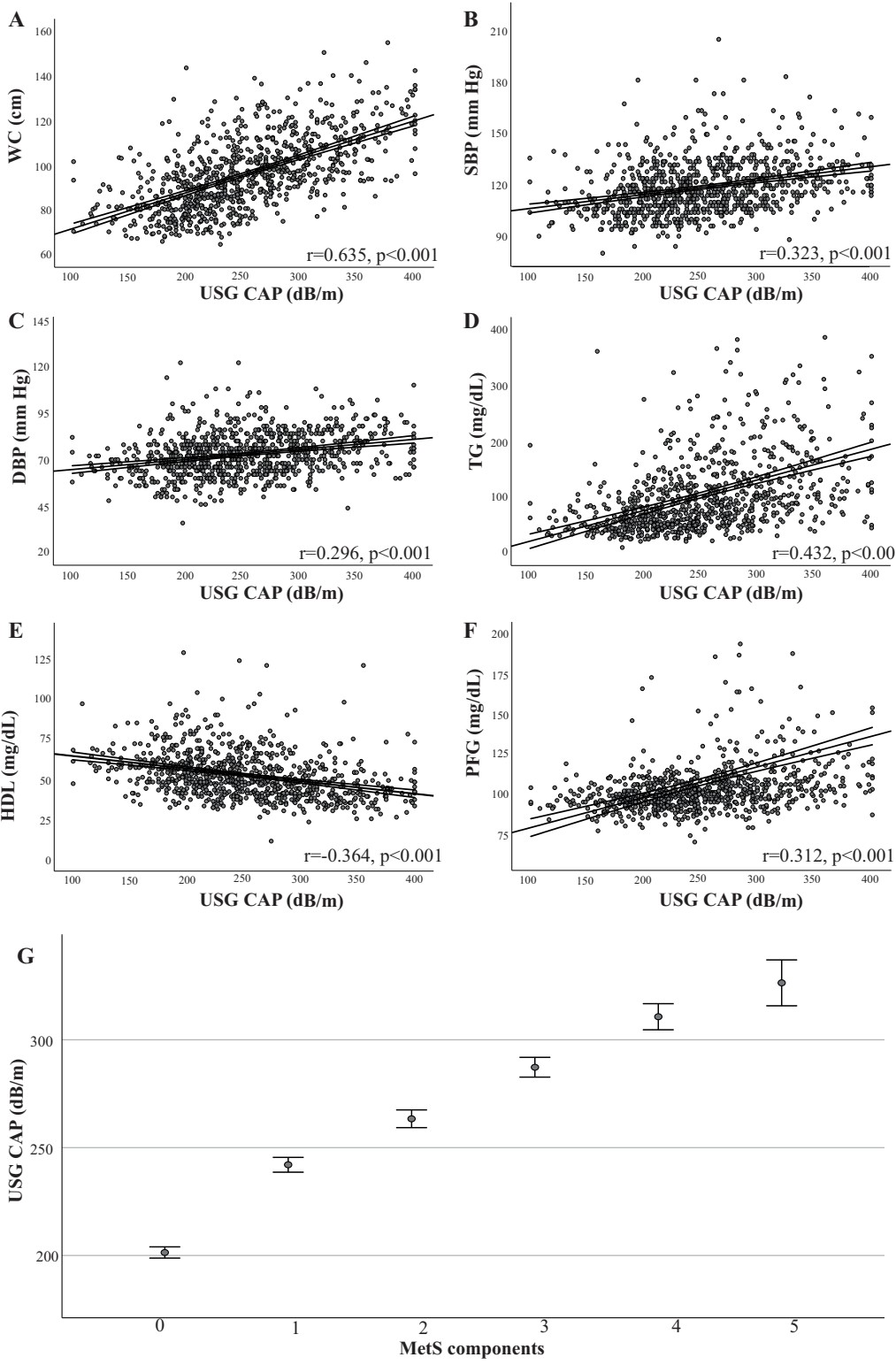

**Fig 4. Association between different components of MetS and controlled attention parameter.** Scattergrams were constructed in which **A** waist conference (WC), **B** systolic blood pressure (SBP), **C** diastolic blood pressure (DBP), **D** triglycerides (TG), **E** high-density lipoprotein (HDL), and **F** fasting plasma glucose (FPG) were compared to liver elastography controlled attenuation parameter (USG CAP) values. Pearson's correlation coefficient was calculated to determine the strength of the association. Lines correspond to the linear fit with 95% confidence intervals. **G** Using the

Harmonizing Definition for MetS, the number of positive categories (0 to 5) was determined. Jockheere-Terpstra test indicated a trend between USG CAP and a number of MetS components (p$_{Jockheere-Terpstra}$ <0.001). Data are shown as mean (dots) and 95% CI (bars).

attributed it to two main causes: 1) the inability of BMI to distinguish between the amount and distribution of lean and fat tissues, and 2) the different metabolic phenotypes of obesity, as determined by body fat distribution and VAT accumulation. Therefore, our results suggest that VATob is a major contributor when considering lean-NALFD. However, this effect almost remained (p = 0.060) for VATob in normal-weight participants when controlling for age and biological sex. Moreover, for SATob, normal-weight participants demonstrated an increased risk for NAFLD when controlling for age and biological sex as well as ethnicity, ALT, HbA1c, and HDL. Biological sex and age were identified as independent predictors for NAFLD [49, 50]. Here, females presented a decreased risk by themselves but was lost when age and other confounders were considered. Numerous studies have shown the association between many clinical and demographical variables and NAFLD [3, 51, 52], in which the results are conflicting. Here, due to the completeness of the data and the chances for multicollinearity, the selected model was made. Nevertheless, the association could be affected by addition of other variables. Therefore, more research is required about specific confounding variables before selecting an optimal set for determining the effect VAT and SAT have on NAFLD development.

Some studies that evaluate the association between SAT and NAFLD have posited that SAT is not relevant for NAFLD development and should not be considered as a risk factor [53]. In contrast, some researchers have reported that SAT can have a reverse longitudinal association with NAFLD causing its regression, whereas others have described a significant association between SAT and NAFLD, independent of VAT [54, 55]. In this study, for the unadjusted model, there was a significant association between SAT and the development of NAFLD and became stronger when other confounding variables were considered. The disparity observed in previous studies has encouraged further research about possible differences in the population, such as ethnicity and nutrition, that could affect metabolic activity of SAT.

Table 6. Linear regression between USG CAP and components of MetS.

| Category | Crude [a] | Model 1 [b] | Model 2 [c] |
|---|---|---|---|
| Components of MetS | | | |
| WC | 2.3 (2.0–2.6), 0.001* | 2.1 (1.9–2.4), 0.001* | 1.9 (1.6–2.2), 0.001* |
| SBP | 1.3 (0.9–1.7), 0.001* | 0.9 (0.5–1.3), 0.001* | 0.7 (0.4–1.0), 0.001* |
| DBP | 1.5 (0.8–2.2), 0.001* | 1.0 (0.4–1.6), 0.003* | 0.7 (0.1–1.4), 0.023* |
| HDL | -1.5 (-1.8–-1.3), 0.001* | -1.6 (-1.8 –-1.4), 0.001* | -1.3 (-1.5 –-1.1), 0.001* |
| TG | 0.33 (0.28–0.38), 0.001* | 0.28 (0.22–0.34), 0.001* | 0.22 (0.15–0.29), 0.001* |
| FPG | 0.67 (0.38–0.96), 0.001* | 0.51 (0.23–0.80), 0.002* | 0.06 (-0.40–0.52), 0.793 |
| Presence of MetS | 60.4 (48.8–72.0), 0.001* | 50.5 (37.1–63.8), 0.001* | 36.6 (24.8–48.5), 0.001* |

Abbreviations: ALT: alanine transaminase; DBP: diastolic blood pressure; FPG: fasting plasma glucose; HbA1c: glycated hemoglobin; MetS Metabolic Syndrome; SBP: systolic blood pressure; TG: triglycerides; WC: waist circumference; and USG CAP: ultrasound with the Controlled Attenuation Parameter.

[a] Values are the beta coefficients (95% confidence interval), and p-value. Values were calculated using the Complex Sample Design General Linear Model. * indicates a significant result (p<0.05, two-tailed).

[b] Model 1 = crude model adjusted for age and sex.

[c] Model 2 = crude model adjusted for age, sex, ethnicity, ALT, and HbA1c|.

**Table 7. The effect MetS and either VATob or SATob have on the risk for NAFLD.**

| Group | Total (cases) [a] | Crude [b] | Model 1 [c] | Model 2 [d] |
|---|---|---|---|---|
| *VATob* | | | | |
| MetS- / VATob- | 22,373,865 (11.5%) | 1.0 (Referent) | 1.0 (Referent) | 1.0 (Referent) |
| MetS- / VATob+ | 9,975,885 (48.9%) | 7.3 (4.4–12.1), <0.001* | 7.4 (4.3–12.6), <0.001* | 6.6 (3.8–11.3), <0.001* |
| MetS+ / VATob- | 2,723,908 (44.9%) | 6.3 (2.6–15.2), <0.001* | 5.6 (2.3–13.5), 0.001* | 5.6 (2.2–14.3), 0.002* |
| MetS+ / VATob+ | 9,326,039 (70.2%) | 18.1 (8.0–41.3), <0.001* | 16.9 (6.4–44.9), <0.001* | 10.3 (4.0–27.0), <0.001* |
| *SATob* | | | | |
| MetS- / SATob- | 18,859,921 (10.5%) | 1.0 (Referent) | 1.0 (Referent) | 1.0 (Referent) |
| MetS- / SATob+ | 13,489,829 (40.5%) | 5.7 (3.3–10.3), <0.001* | 8.4 (4.1–16.9), <0.001* | 7.4 (3.4–15.9), <0.001* |
| MetS+ / SATob- | 3,356,503 (54.9%) | 10.3 (4.8–22.4), <0.001* | 5.9 (2.5–14.0), 0.001* | 5.0 (2.0–12.8), 0.002* |
| MetS+ / SATob+ | 8,693,444 (68.2%) | 18.3 (8.0–41.9), <0.001* | 17.7 (7.3–42.9), <0.001* | 11.2 (4.8–26.4), <0.001* |

Abbreviations: MetS: Metabolic Syndrome; SATob: Subcutaneous fat obesity; and VATob: Visceral fat obesity.

[a] Total weighted number of participants for each group (percent positive for NAFLD).

[b] Values are odds ratios (95% confidence interval), and p-value. Odds ratios were determined by comparing each group to the referent group for NAFLD using Complex Samples Design Logistic Regression. * Indicates a significant result (p<0.05, two-tailed).

[c] Model 1 = crude model adjusted for age and sex

[d] Model 2 = crude model adjusted for age, sex, race, alanine transaminase, and HbA1c.

The prevalence of NAFLD has been increasing in the past decades; however, as stated by Bonacini *et al.* and as shown in this study, this increase has an ethnic disparity [56]. According to a recent meta-analysis, the overall prevalence of NAFLD worldwide is 32.4% [1], in which Hispanics and Non-Hispanic Blacks presented with the highest and lowest proportion of the population, respectively [51]. This corresponds with our study, which posits possible ethnic-based mechanisms that influence the physiopathology of NAFLD. When compared to Non-Hispanic Whites, Mexican Americans presented a significant risk for NAFLD, whereas none of other ethnicities did. Moreover, Mexican Americans had the largest proportion of level 3 steatosis, which could lead to higher rates of disease progression and fibrosis. However, when only VATob was considered, Non-Hispanic Whites and Non-Hispanic Blacks presented with the highest risk and Mexican Americans did not. With respect to SATob, again, Non-Hispanic Whites and Non-Hispanic Blacks as well as other Hispanics presented with the higher risk, whereas Mexican Americans and Non-Hispanic Asians presented with the lower risk. The mechanism is not fully understood, but could be credited to variations in metabolic pheno-types, genetic predisposition, as well as cultural and socioeconomical factors [52]. For example, regarding genetic predisposition, the variation in the risk allele of the *patatin-like phospholi-pase domain-containing protein* 3 (PNPLA3), a gene that confers susceptibility for NAFLD, is more frequent in Hispanics (49%) followed by non-Hispanic Whites (23%), and Non-Hispanic Blacks (17%), which can account for the increased risk in the Hispanic population [57, 58]. Nevertheless, the ethnic disparity cannot be attributed only to genetic factors. It has been shown that risks differ between Japanese subjects born and raised in Japan and Japanese Americans, as well as in Africans from Nigeria and Non-Hispanic Blacks, implicating socio-economic and lifestyle factors [59, 60]. The main lifestyle condition associated with NAFLD is dietary patterns and lack of physical activity [61]. Overall, increased intake of fructose, choles-terol, and foods high in saturated fats predisposes subjects to NAFLD [62, 63]. The Western dietary pattern and eating habits, characterized by containing large amounts of red meat, pro-cessed meat, fried foods, and glucose-rich soft drink consumption have been shown to increase liver fat and were strongly associated with higher values of elastography measures (OR = 4.21)

[64]. On the other hand, a traditional Chinese diet—vegetable rich, rich in low-fat dairy, nuts, fruit, coffee, and tea—was protective (OR = 0.26) [64]. These different dietary patterns were shown to affect VAT and SAT development, altering metabolic phenotypes [65, 66]. Nevertheless, when the severity of NAFLD was considered, a recent meta-analysis conducted by Rich *et al.* stated there was no significant difference in severity of steatosis among Hispanics, Non-Hispanic Whites, and Non-Hispanic Blacks [51]. Overall, this could indicate that ethnicity is a determinant for the different metabolic phenotypes that abdominal fat depots exhibit, affecting NAFLD prevalence.

Lastly, regarding the different metabolic phenotypes for the study population, MetS was evaluated as a sub-analysis. NAFLD is recognized as the hepatic manifestation of MetS [16]. In MetS individuals, NAFLD's prevalence is 43%, which increases with the number of MetS components, reaching 63% in subjects with 5 components [17]. Here, the prevalence of MetS in NAFLD participants was 51.1%, which is in accordance with other studies [16]. Here, each component of MetS was associated with USG CAP, as well as there was a consistent increase in USG CAP with each increase in the number of MetS components. Jinjuvadia *et al.* demonstrated individual components of MetS are independent predictors for the development of NAFLD, except for hypertension [17]. This is also supported by other studies by Chon *et al.* [67] and Huang *et al.* [68], in which USG CAP scores were positively associated with T2D markers for insulin resistance and metabolic dysfunction With respect to different types of abdominal fat accumulation affecting the interaction between MetS and NAFLD, we show that participants with MetS and VATob were found to be 18-times more likely to develop NAFLD when compared to the control, which is more than participant with MetS only (OR = 6.3). This value was higher than another study, in which a 11-fold increased risk was observed due to MetS [17]; however, this study did not consider the type of abdominal fat. Here, for SATob, similar results were observed, except for participants with MetS only, their ORs were higher (OR = 10.3). Nevertheless, when adjusted by the selected variables, independent if SATob or VATob was being assessed, the results were comparable. When using BMI to determine obesity, similar results are shown [39]. However, VAT and SAT have different metabolic activities and functions [10] that should affect the effect due to MetS on NAFLD risk. This is an example of the problem between evaluating obesity, when considering the type of fat. To properly address the presence of VATob and SATob, eight groups would be needed to determine the effect; and due to the sample size, this analysis could not be done.

This study has a few limitations. First, this is a cross sectional study and causation cannot be concluded; therefore, the results should be assessed cautiously. Second, statistical adjustments were not made for insulin resistance or use of female hormones; however, alcohol consumption and other possible causes of liver disease were excluded. Third, this analysis was performed on participants older than 17 years and the conclusion should only be used for an adult cohort. Fourth, for NAFLD, as well as other complication, associations between independent variables and the outcome are affect by the inclusion of proper confounder. Here, may potential confounders were excluded due to missing data. Lastly, ultrasound is an operator dependent technique, and these results were not compared against the gold standard for NAFLD—liver biopsy. Shalimar *et al.* suggested using a BMI adjusted USG CAP thresholds for NAFLD diagnosis [69]. USG CAP scores have demonstrated sufficient sensibility and specificity to identify NAFLD cases and BMI adjustments have not been added the diagnostic consensus.

In conclusion, TAFA, VAT, and SAT were positively associated with USG CAP values, supporting that abdominal fat contributes to the development of NAFLD. Additionally, ethnicity plays an important role in the risk and development of NAFLD, but this association is altered by VATob and SATob. Therefore, futures studies need to consider how type of abdominal fat

depots and ethnicity promote NAFLD. Lastly, the type of abdominal fat depots did affect the association between MetS and NAFLD.

## Supporting information

**S1 Checklist. Strengthening the Reporting of Observational Studies in Epidemiology (STROBE) guideline.**
(DOC)

**S1 Fig. Flowchart for the selection of study participants.**
(PDF)

**S1 Table. Comparison of Pearson correlation coefficients for TAFA, VAT, SAT, and BMI with liver steatosis, as measured with hepatic ultrasound with the controlled attenuation parameter.**
(PDF)

**S2 Table. Predictability of type of abdominal fat depots and BMI for NAFLD as determined using receiver operating characteristic curve analysis.**
(PDF)

## Acknowledgments

The authors would like to express their gratitude to Mtro. Alfredo Avendaño Arenaza, Director of the BUAP Central University Library, and Mtro. Ricardo Villegas Tovar, Coordinator of Scientific Production and International Visibility, at the Benemérita Universidad Autónoma de Puebla. The authors would also like to thank the participants and the NHANES staff for their valuable contributions. The authors assume full responsibility for analyses and interpretation of these data.

## Author Contributions

**Conceptualization:** Rebeca Garazi Elguezabal Rodelo, Leonardo M. Porchia, M. Elba Gonzalez-Mejia.

**Data curation:** Rebeca Garazi Elguezabal Rodelo, Leonardo M. Porchia.

**Formal analysis:** Rebeca Garazi Elguezabal Rodelo, Leonardo M. Porchia.

**Funding acquisition:** M. Elba Gonzalez-Mejia.

**Methodology:** Rebeca Garazi Elguezabal Rodelo, Leonardo M. Porchia, Esther López-Bayghen, M. Elba Gonzalez-Mejia.

**Project administration:** M. Elba Gonzalez-Mejia.

**Resources:** Enrique Torres-Rasgado, Esther López-Bayghen.

**Supervision:** Enrique Torres-Rasgado, Esther López-Bayghen, M. Elba Gonzalez-Mejia.

**Visualization:** M. Elba Gonzalez-Mejia.

**Writing – original draft:** Rebeca Garazi Elguezabal Rodelo, Leonardo M. Porchia.

**Writing – review & editing:** Enrique Torres-Rasgado, Esther López-Bayghen, M. Elba Gonzalez-Mejia.

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
