## [Decision Letter · Decision Letter 0]

10 Dec 2023

PONE-D-23-40506The effect of different types of abdominal fat on Non-alcoholic fatty liver diseasePLOS ONE

Dear Dr. López-Bayghen,

Thank you for submitting your manuscript to PLOS ONE. After careful consideration, we feel that it has merit but does not fully meet PLOS ONE’s publication criteria as it currently stands. Therefore, we invite you to submit a revised version of the manuscript that addresses the points raised by the reviewer.

We look forward to receiving your revised manuscript.

Kind regards,

Matias A Avila, Ph.D.

Academic Editor

PLOS ONE

Journal Requirements:

Reviewers' comments:

Reviewer's Responses to Questions

**Comments to the Author**

1. Is the manuscript technically sound, and do the data support the conclusions?

Reviewer #1: Partly

2. Has the statistical analysis been performed appropriately and rigorously? 

Reviewer #1: No

3. Have the authors made all data underlying the findings in their manuscript fully available?

Reviewer #1: No

4. Is the manuscript presented in an intelligible fashion and written in standard English?

Reviewer #1: Yes

5. Review Comments to the Author

Reviewer #1: The article titled "The Effect of Different Types of Abdominal Fat on Non-alcoholic Fatty Liver Disease" (PONE-D-23-40506) assesses the association between fat mass from various depots and NAFLD in the NHANES dataset. This article requires a comprehensive review: the analyses are moderately rigorous, and the reporting of results needs refinement. The discussion should encompass all previous literature with a similar approach and/or outcomes. Consequently, there are several aspects that need improvement.

Major concerns:

1. The study lacks a comprehensive review of relevant literature with a similar approach and/or outcomes. It is crucial to integrate findings from studies that share common themes or investigate comparable aspects of NAFLD. Consider incorporating insights from Huang et al. (PMID: 35173677), Huh et al. (PMID: 27923446), Lin et al. (PMID: 31653028), Chon et al. (PMID: 27189281), Shalimar et al. (PMID: 32185692), and Shen et al. (PMID: 24782622) to provide a more robust context for the current study's results. By doing so, the discussion will benefit from a broader understanding of the field and establish connections with existing research.

2. The article's analyses are deemed moderately rigorous, suggesting a need for restructuring both the methodology and the presentation of results. To enhance clarity and precision, refer to the exemplary approach taken by Huang et al. (PMID: 35173677) in their article. Particularly, examine Table 3 in their publication, which serves as a clear illustration of how to effectively report odds ratio (OR) results. Adopting similar reporting practices will contribute to the overall methodological rigor and comprehensibility of the study.

3. Drawing inspiration from Huh et al. (PMID: 27923446), who explored the interplay of obesity and metabolic health on hepatic steatosis and fibrosis, especially as defined by CAP, can offer valuable insights. Refer to Table 3 in their study to understand how to present results of this nature. Additionally, consulting the approaches taken by Lin et al. (PMID: 31653028), Chon et al. (PMID: 27189281), Shalimar et al. (PMID: 32185692), and Shen et al. (PMID: 24782622) will provide guidance on explaining adjustment methodologies and variable selection in a manner that ensures transparency and replicability.

4. To strengthen the statistical robustness of the study, consider employing at least three models in the analyses. These models should include an unadjusted one, a second model incorporating simple covariates such as age and sex, and a third model incorporating all variables that exhibit statistically significant differences between groups. This multi-model approach is particularly vital in association studies utilizing logistic regression, as it evaluates relationships between predictor variables and binary outcomes. Inadequate adjustments may lead to weak conclusions, emphasizing the importance of a thorough and meticulous analytical strategy.

5. Expanding the scope of the discussion to encompass additional studies examining the association between visceral adipose tissue and NAFLD is essential. Numerous studies, both longitudinal and transversal, have explored this relationship (refer to the review by Seaw et al., https://doi.org/10.3390/livers3030033). The authors are encouraged to thoroughly review these articles, incorporate their findings into the discussion, and draw comparisons with their own results when appropriately executed. Notably, within the cited review by Seaw et al., there are at least five articles (Jeong et al., Lee & Kuk et al., Simon et al., Hu et al., and Chiyanika et al.) that did not stratify participants by gender, while four others (Damaso et al., Kim et al., Igarashi et al., and Lee et al.) did. Additionally, two studies (Kure et al. and Baek et al.) categorized participants by visceral adipose tissue (VAT) mass, and Liu et al. (PMID: 36760579) explored the association between VAT and NAFLD in populations stratified by glucose tolerance. Despite the existence of similar data in the literature, this paper has the potential to contribute significantly by comparing results across diverse populations or highlighting variations in other terms. The review by Seaw et al. underscores the scarcity of studies conducted in non-Asian populations, emphasizing the need for a broader demographic representation.

6. It is recommended that the authors undertake an analysis of the association between metabolic syndrome and its components with NAFLD within the present manuscript, providing additional information on this crucial aspect. Furthermore, considering an examination of fat mass in different depots, BMI, and metabolic syndrome with controlled attenuation parameter (CAP) levels would enhance the depth of the study. These additional analyses would contribute valuable insights into the multifaceted interplay between various factors and NAFLD, enriching the overall scientific contribution of the manuscript.

Minor Changes:

1. The title requires refinement to provide a more precise and informative statement. It should explicitly mention the main outcome, the type of analysis (association/relationship), and the data source (NHANES cross-sectional study/transversal).

2. Ensure that statistics accompany the plots in Figure 1, enhancing the clarity and completeness of the visual representations.

3. Consider addressing the absence of a sub-analysis involving adolescents and/or children in the manuscript. This could potentially be one of the pioneering articles to explore such an analysis if conducted accurately. Providing insights into this demographic group could offer valuable contributions to the existing literature.

6. PLOS authors have the option to publish the peer review history of their article (what does this mean?). If published, this will include your full peer review and any attached files.

Reviewer #1: **Yes: **Amaya Lopez-Pascual

---

## [Author Response · Author response to Decision Letter 0]

23 Jan 2024

Dear Editor, 

We want to thank the Reviewers for this evaluation of our manuscript, "The effect of different types of abdominal fat on Non-alcoholic fatty liver disease" (Manuscript ID: PONE-D-23-40506). We have considered the comments and have taken the appropriate actions. For each of the comments, we have written a reply below.

figures

Journal Requirements:

Response: We have followed the example files and made sure the naming of the documents is correct.

Response: We have placed the data in the Harvard Dataverse repository. The DOI is: https://doi.org/10.7910/DVN/UW4EER. This information was also added to the text.

Reviewer #1: The article titled "The Effect of Different Types of Abdominal Fat on Non-alcoholic Fatty Liver Disease" (PONE-D-23-40506) assesses the association between fat mass from various depots and NAFLD in the NHANES dataset. This article requires a comprehensive review: the analyses are moderately rigorous, and the reporting of results needs refinement. The discussion should encompass all previous literature with a similar approach and/or outcomes. Consequently, there are several aspects that need improvement.

Major concerns:

1. The study lacks a comprehensive review of relevant literature with a similar approach and/or outcomes. It is crucial to integrate findings from studies that share common themes or investigate comparable aspects of NAFLD. Consider incorporating insights from Huang et al. (PMID: 35173677), Huh et al. (PMID: 27923446), Lin et al. (PMID: 31653028), Chon et al. (PMID: 27189281), Shalimar et al. (PMID: 32185692), and Shen et al. (PMID: 24782622) to provide a more robust context for the current study's results. By doing so, the discussion will benefit from a broader understanding of the field and establish connections with existing research.

Response: We agree with the reviewer and have added the indicated articles, as well as others we found, to the discussion. 

2. The article's analyses are deemed moderately rigorous, suggesting a need for restructuring both the methodology and the presentation of results. To enhance clarity and precision, refer to the exemplary approach taken by Huang et al. (PMID: 35173677) in their article. Particularly, examine Table 3 in their publication, which serves as a clear illustration of how to effectively report odds ratio (OR) results. Adopting similar reporting practices will contribute to the overall methodological rigor and comprehensibility of the study.

Response: We have adopted some of the methodology presented in the Huang et al. study. We converted all tables to be similar to the tables present in their study as well as the Huh et al. study. We wish to point out that the Huang et al. study used SRS, whereas the NHANES dataset was from a complex samples design. Therefore, we could only use certain tests under certain assumptions.

Concerning reporting odds ratios, we have added some of the details demonstrated by Huang et al. study in Table 3 as well as the Huh et al. study; however, we choose not to show the referent group (the non-obese group) for each comparison as its own line, as it would make the table long and redundant. We have published tables like this before. Nevertheless, we have corrected Table 2’s title to indicate the referent is the non-obese group.

3. Drawing inspiration from Huh et al. (PMID: 27923446), who explored the interplay of obesity and metabolic health on hepatic steatosis and fibrosis, especially as defined by CAP, can offer valuable insights. Refer to Table 3 in their study to understand how to present results of this nature. Additionally, consulting the approaches taken by Lin et al. (PMID: 31653028), Chon et al. (PMID: 27189281), Shalimar et al. (PMID: 32185692), and Shen et al. (PMID: 24782622) will provide guidance on explaining adjustment methodologies and variable selection in a manner that ensures transparency and replicability.

Response: With respect to adjusting associations (linear or logistic regression), there are two main ways, as pointed out by Magdalena Szumilas (PMID: 20842279) and Mohamad Amin Pourhoseingholi (PMID: 24834204): 1) (the more typical) adding the confounding variable as a term to the regression equation; and 2) stratification by the confounding variable. We agree that we should have at least adjusted by biological sex as well as age and now have provided this. However, we tried to control for central obesity by stratifying BMI into its categories. When deciding which variables should be adjusted, as stated by Bursac et al., “Some methodologists suggest including all clinical and other relevant variables in the model regardless of their significance to control for confounding. This approach, however, can lead to numerically unstable estimates and large standard errors” (PMID: 19087314). Thus, related variables can lead to a misinterpreted association that lose a significant result due to the high level of correlation between the variables, as well as any causal relationship between 2 variables should be taken into consideration, as the dependent variable should be included upon knowing how the causal effect can affect the association. Here, for other variables, their inclusion into the regression model was based on statistical significance and to minimize redundant effects, such as excluding waist circumference and other measures of central obesity due to the stratified analysis with BMI.

With respect to Huh et al., we have followed their methodology and have made the appropriate corrections to the methods as well as the results. Nonetheless, we wish to point out in their Table 3 that the order of the middle to categories (MUNO and MHO) could be switched and affect the trend test, as there is no agreement or evidence to undoubtably support MUNO is less healthy than MHO. It would have been better to assess if an additive biological interaction occurs. Huang et al. trend test was performed appropriately.

4. To strengthen the statistical robustness of the study, consider employing at least three models in the analyses. These models should include an unadjusted one, a second model incorporating simple covariates such as age and sex, and a third one incorporating all variables exhibiting statistically significant differences between groups. This multi-model approach is particularly vital in association studies utilizing logistic regression, as it evaluates relationships between predictor variables and binary outcomes. Inadequate adjustments may lead to weak conclusions, emphasizing the importance of a thorough and meticulous analytical strategy.

Response: We agree with the author and have provided the 3 models that were requested. 

Just so the reviewer knows, the NHANES does not collect all data for each participant; therefore, some participants were missing data for non-key variables. This data was initially reported, but we should have indicated that many participants were missing this data. Here, if >5% of the cohort was missing data for each potential variable, then the variable was not reported and removed. For the main results, no data was missing for each variable (CAP and types of fats). However, when adjusting or selecting variables to be adjusted, we determined that if the sample size did not decrease by >5% as well as the portions between the independent and dependent variables remained constant (<1% change), then the variable could be included. These adjusted models were not performed/utilized if these assumptions were violated. This information was added to the Methods and Results sections. We provided a table examining the potential associations between certain confounding variables and NAFLD. Using the results from this table, we developed our selected adjusted model, similar to Lin et al. All tables that an association between the independent and dependent variables were assessed were also adjusted by the 2 models. 

5. Expanding the scope of the discussion to encompass additional studies examining the association between visceral adipose tissue and NAFLD is essential. Numerous studies, both longitudinal and transversal, have explored this relationship (refer to the review by Seaw et al., https://doi.org/10.3390/livers3030033). The authors are encouraged to thoroughly review these articles, incorporate their findings into the discussion, and draw comparisons with their own results when appropriately executed. Notably, within the cited review by Seaw et al., there are at least five articles (Jeong et al., Lee & Kuk et al., Simon et al., Hu et al., and Chiyanika et al.) that did not stratify participants by gender, while four others (Damaso et al., Kim et al., Igarashi et al., and Lee et al.) did. Additionally, two studies (Kure et al. and Baek et al.) categorized participants by visceral adipose tissue (VAT) mass, and Liu et al. (PMID: 36760579) explored the association between VAT and NAFLD in populations stratified by glucose tolerance. Despite the existence of similar data in the literature, this paper has the potential to contribute significantly by comparing results across diverse populations or highlighting variations in other terms. The review by Seaw et al. underscores the scarcity of studies conducted in non-Asian populations, emphasizing the need for a broader demographic representation.

Response: We agree with the reviewer and have added the requested information to the discussion.

6. It is recommended that the authors undertake an analysis of the association between metabolic syndrome and its components with NAFLD within the present manuscript, providing additional information on this crucial aspect. Furthermore, considering an examination of fat mass in different depots, BMI, and metabolic syndrome with controlled attenuation parameter (CAP) levels would enhance the depth of the study. These additional analyses would contribute valuable insights into the multifaceted interplay between various factors and NAFLD, enriching the overall scientific contribution of the manuscript.

Response: As mentioned above, the NHANES does not collect all data for each participant. This does present a problem, in which ~50% of the sample is missing data for triglycerides and fasting plasma glucose. To resolve this issue, this analysis was performed as a sub-analysis for the components of Metabolic Syndrome. We indicate that the overall sample size has decreased in the results. Lastly, keeping with the original focus of the manuscript, we tested the effect VAT and SAT have on the association between MetS and NAFLD. We agree that this analysis is fascinating, and we are happy the reviewer suggested it. As pointed out in the discussion, a more comprehensive analysis needs to be designed, which we are planning. And as per Minor Changes #3, we hope to be able to include adolescents. 

Minor Changes:

1. The title requires refinement to provide a more precise and informative statement. It should explicitly mention the main outcome, the type of analysis (association/relationship), and the data source (NHANES cross-sectional study/transversal).

Response: We have corrected the title to have these components.

2. Ensure that statistics accompany the plots in Figure 1, enhancing the clarity and completeness of the visual representations.

Response: We have added a linear regression line with the 95% confidence interval as well as the Pearson correlation coefficient and p-value to all scatterplots. 

3. Consider addressing the absence of a sub-analysis involving adolescents and/or children in the manuscript. This could potentially be one of the pioneering articles to explore such an analysis if conducted accurately. Providing insights into this demographic group could offer valuable contributions to the existing literature.

Response: We agree that this information would be interesting. Children are outside our typical study age, and due to the time limit for this analysis, we have decided not to include it here. Moreover, we wish to see what the reviewer thinks about the updated and additional analyses. Nevertheless, we are going to perform this analysis in a future study. This information was added as a limitation.

---

## [Editor Report · Decision Letter 1]

30 Jan 2024

Visceral and subcutaneous abdominal fat is associated with Non-alcoholic fatty liver disease while augmenting Metabolic Syndrome’s effect on Non-alcoholic fatty liver disease: A cross-sectional study of NHANES 2017-2018

PONE-D-23-40506R1

Dear Dr. López-Bayghen,

We’re pleased to inform you that your manuscript has been judged scientifically suitable for publication and will be formally accepted for publication once it meets all outstanding technical requirements.

Kind regards,

Matias A Avila, Ph.D.

Academic Editor

PLOS ONE
---

## [Editor Report · Acceptance letter]

13 Feb 2024

PONE-D-23-40506R1 

PLOS ONE

Dear Dr. López-Bayghen, 

I'm pleased to inform you that your manuscript has been deemed suitable for publication in PLOS ONE. Congratulations! Your manuscript is now being handed over to our production team.

Kind regards, 

on behalf of

Dr Matias A Avila 

Academic Editor

PLOS ONE